# Effects of Weather and Environmental Factors on the Seasonal Prevalence of Foodborne Viruses in Irrigation Waters in Gyeonggi Province, Korea

**DOI:** 10.3390/microorganisms8081224

**Published:** 2020-08-11

**Authors:** Zhaoqi Wang, Hansaem Shin, Soontag Jung, Daseul Yeo, Hyunkyung Park, Sangah Shin, Dong Joo Seo, Ki Hwan Park, Changsun Choi

**Affiliations:** 1Department of Food and Nutrition, Chung-Ang University, Gyeonggi-do 17546, Korea; wzqlayumi8@gmail.com (Z.W.); dukot@naver.com (H.S.); amazing2257@gmail.com (S.J.); da159951@gmail.com (D.Y.); opp6049@hanmail.net (H.P.); ivory8320@cau.ac.kr (S.S.); 2Department of Food and Nutrition, Gwangju University, Gwangju 61743, Korea; sdj0118@gwangju.ac.kr; 3Department of Food Science and Technology, Chung-Ang University, 4726, Gyeonggi-do 17546, Korea; khpark@cau.ac.kr

**Keywords:** norovirus, adenovirus, reservoir, contamination, precipitation, drought

## Abstract

This study aimed to investigate the prevalence of foodborne viruses in reservoirs (an important resource of irrigation water) and its correlation with environmental and weather factors. From May 2017 to November 2018, we visited ten reservoirs and a river in the Anseong region of South Korea and collected a total of 192 samples in accordance with the environment protection agency guidelines. We recorded the weather factors (temperature, humidity, and accumulated precipitation) and investigated the surrounding environment factors (livestock, fishing site, the catchment area of reservoirs, etc.). Our research results show that from the river and reservoirs, the detection rates of human norovirus GII, adenovirus, rotavirus, human norovirus GI, and astrovirus were 27.1, 10.4, 10.4, 4.16, and 3.1%, respectively. Their viral load ranged from −1.48 to 1.55 log_10_ genome copies/l. However, hepatitis A virus was not detected in any irrigation water sample. Although no sampling was performed in winter, foodborne viruses and male-specific coliphages were frequently found during spring (40.78%) and autumn (39.47%). Interestingly, the significant correlation between the accumulative precipitation and the number of detected norovirus and adenovirus was confirmed by linear regression analysis. Furthermore, when the accumulative precipitation ranged from 20 to 60 mm, it significantly affected the viral load and prevalence. Among the environmental factors, recreational facilities such as fishing sites and bungalow fishing spots were identified as contamination sources by correlation analysis. Our research results confirmed the correlations between environmental contamination factors in the reservoir and weather factors with the prevalence of foodborne viruses in the reservoir. These facilitates the assessment of potential foodborne virus contamination during crop irrigation. In addition, predictive models including environmental and weather factors should be developed for monitoring and controlling the safety of irrigation waters in reservoirs.

## 1. Introduction

Climate change has been shown to affect ecosystems worldwide [1]. The Intergovernmental Panel on Climate Change (IPCC) has reported that extreme weather events, including heavy rains and droughts, are likely to occur at higher frequencies in several countries. In particular, severe droughts are likely to be one of the worst natural disasters resulting from climate change [1]. Although Korea is equipped with excellent water supply facilities, the recent increase in the occurrence of regional droughts is a matter of concern [2]. Four global climate models (GCMs) have predicted that drought hazard due to climate change is likely to be severe in the eastern and central regions of Korea in contrast to the southern region [3]. A severe drought impacted the Korean peninsula in 2017. Consequently, agricultural farms, including those in Gyeonggi province, experienced severe drought for several months [4,5].

Reservoirs play an important role in supplying irrigation water, especially during periods of frequent water utilization. South Korea has a total of 3,049 reservoirs, which not only serve as a source of irrigation water, but also supply water during droughts [2]. Complex factors, including weather and local environment, can affect the water quality of reservoirs. Urban sewage, manure fertilizers, livestock farms, and industrial production have been shown to contaminate irrigation water [6,7]. Among weather factors, heavy rains and floods can significantly affect the severity of irrigation water contamination [8,9]. Droughts have also been shown to result in 100-fold higher concentrations of pathogens in surface waters, thus altering the water quality [10]. Humidity, temperature, and sunlight can also affect the survival of pathogens [11,12].

The contamination of irrigation water with bacteria, fungi, viruses, and protozoa is detrimental to public health [10,13]. Utilization of contaminated irrigation water can not only cause pathogens to attach to the cut surfaces of leafy greens through the stoma, but also result in their accumulation in fresh produce through water absorption by roots [14,15,16]. Agricultural farms obtain irrigation water from various sources, such as reservoirs, rivers, and groundwater; they are thus susceptible to pathogenic contamination of fresh produce and fruits [10,17,18]. In recent decades, contaminated fresh produce and fruits have been shown to be associated with viral gastroenteritis [19]. These results suggest that irrigation water safety is closely associated with food safety. Therefore, it is crucial to not only investigate the contamination sources, but also determine the relationships between prevalence of foodborne viruses in reservoirs and weather and environmental factors.

Foodborne viral infections exert a significant economic burden globally [20]. Human norovirus (HuNoV), human adenovirus (AdenoV), human astrovirus (AstroV), and human rotavirus (RotaV) frequently cause acute viral gastroenteritis characterized by abdominal cramps, nausea, vomiting, diarrhea, and fever. Hepatitis A virus (HAV) and hepatitis E virus (HEV) are associated with acute hepatitis and jaundice [21,22,23]. The first reported outbreaks of foodborne gastroenteritis in school children in Korea occurred in 2004 and were associated with HuNoV [24]. Subsequently, 50 viral foodborne outbreaks and 1,211 patients have been reported annually (2012 to 2018) by the Ministry of Food and Drug Safety in Korea [25]. Furthermore, Korea has reported frequent HAV, RotaV, AdenoV, and AstroV outbreaks [26,27,28]. Although groundwater contamination by HuNoV and HAV was confirmed as the cause of several outbreaks in Korea [29,30], the prevalence of foodborne viruses in reservoirs in Korea has not been previously reported.

Therefore, it is important to examine the microbial safety of reservoirs supplying irrigation water to reduce the risk of viral contamination in fresh produce. This study aimed to monitor pathogenic foodborne viruses (HuNoV, AdenoV, AstroV, HAV, and RotaV) and indicator microorganisms (including male-specific coliphages) in reservoirs in South Korea for a duration of 18 months. Furthermore, the relationships between the prevalence of foodborne viruses in reservoirs and various environmental and weather factors were analyzed.

## 2. Materials and Methods

### 2.1. Sampling Sites and Sample Collection

Anseong is the city surrounded with many agricultural farms. The population of Anseong city is 184,287. This study selected ten man-made reservoirs to supply irrigation water to the Anseong region, Gyeonggi province of South Korea (Figure 1). The Anseong river (site K; characterized by the downstream merging of sites C and D) was also sampled. Sites A, F, and G are small reservoirs located in a flat area surrounded by arable farms and villages. Site D is the large reservoir with a broad catchment area, poundage, and high forest coverage. Sites B and E are located near a golf course with high forest coverage. Many bungalow fishing spots, and recreational facilities were present in sites C, H, I, and J.

A total of 192 irrigation water samples were collected every two months from May 2017 to November 2018 and subsequently examined for viruses (96 samples) and indicator microorganisms (96 samples). However, sampling was not conducted from January 2018 to February 2018 due to surface freezing of the reservoirs. Samples were not collected from the Anseong River (site K) in March 2018 due to a temporary ban on access to this area to prevent the regional outbreak of avian influenza. The surface irrigation water samples were collected following the Environment Protection Agency (EPA) guidelines [31]. Each sampler comprised a flow meter, a pressure gauge, tubing, a cartridge filter housing, and Koshin Hidels Pump (Model: SE-25L) [32]. Approximately 300−350 L of water samples were filtered (10 L/min) using a pre-filter (FROM-10 Deluxe Sediment Filter Cartridge Set). Subsequently, viral particles were concentrated on a 1-MDS electropositive cartridge filter (10 micron, 3M, USA) [32]. Thereafter, the pre-filter was discarded and the secondary 1-MDS filter was either retained in the cartridge housing for immediate estimation of viral concentrations or stored at 4 °C. For indicator microorganisms detection, a 250 mL sterilized water sampling bottle was utilized for irrigation water collection directly from the reservoirs and river. Then, refrigerated and shipped to the laboratory. Samples were analyzed within 24 h after collection.

### 2.2. Virus Concentrations and RNA Extraction

Virus concentrations were determined by the virus adsorption-elution (VIR-AD-EL) technique from the previous study [32]. Initially, 10 µL of MNV-1 as an internal process control was added in 0.5 L of a beef extract solution (1.5% beef extract, 0.05 M glycine, pH = 9.5) to assure the virus recovery. Subsequently, the cartridge housing of each sample was filled with 0.5 L of a beef extract solution by using a peristaltic pump and soaked for 5 min. The above-mentioned process was repeated once. The eluates were adjusted to a pH of 3.5 by using 1 M hydrochloric acid (HCl). Subsequently, the elutes were centrifuged at 2500× *g* at 4 °C for 15 min. Thereafter, the supernatant was removed, and the precipitate was resuspended in 30 mL of a 0.15 M sodium phosphate buffer (Na_2_HPO_4_, pH = 9.0~9.5: Sigma-Aldrich, St. Louis, MO, USA). The supernatant was centrifuged (Beckerman Coulter, Avanti J-E centrifuge, Indianapolis, IN, USA) at 7000x *g* at 4 °C for 10 min and transferred to a new collection tube. The pH was adjusted to 7.0–7.5 using 1 M HCl (Sigma-Aldrich, Billerica MA, USA, CAS Number: 7647-01-0). Then, the concentrated 30 mL water samples were filtered through a 0.45-μm-pore-size syringe filter (Biofact, Daejeon, Korea, Cat. No. BSS45-CA25) to prevent microbial contamination. Ten mL of the filtered samples were concentrated again to 1 mL using Vivaspin 20 (Sartorius, Göttingen, Germany) with a molecular-weight cutoff of 10 k Da. Furthermore, RNA extraction was performed for a 500 μL aliquot of the concentrated sample according to manufacturer’s instructions (RNeasy mini kit, Qiagen, Valencia, CA, USA). The total nucleic acid was eluted with 70 μL of RNase-free water. The extracted nucleic acids were immediately analyzed, and the leftover solution was stored at −80 °C.

### 2.3. Real-Time Reverse Transcription-Quantitative Polymerase Chain Reaction (Real-Time RT-qPCR)

Primers and probes utilized for the seven viruses (HuNoV GI, HuNoV GII, HAV, RotaV, AstroV, AdenoV, and murine norovirus (MNV-1)) and the amplification conditions are listed in Table 1. Real-time qPCR and real-time RT-qPCR were performed using the Takara TP800 Thermal Cycler Dice™ real-time system and Bio-Rad CFX96^TM^ C1000 Touch Thermal Cycler real-time system, respectively. Viral genome was amplified in a 25 µL reaction mixture containing 5 µL of each sample RNA, 12.5 µL of a 2 × QuantiTect Probe RT-PCR Master Mix (QIAGEN, Germany), 0.25 µL of a QuantiTect RT Mix, 500 nM of a forward and reverse primer, and 250 nM of probe. Quantification was performed by using the commercial quantitative synthetic viral RNA (HuNoV GI, ATCC^®^ VR-3234SD™; GII, ATCC^®^ VR-3235SD™; HAV, ATCC^®^ VR-3257SD™; RotaV, ATCC^®^ VR-2018DQ™; AstroV, ATCC^®^ VR-3238SD™; MNV, ATCC^®^ VR-3255SD™). Quantitative DNA fragment of AdenoV was synthesized using Integrated DNA Technologies (IDT, Coralville, IA) [33]. The calculation of virus recovery by using of MNV-1 in this study was based on the previous study [34]. The sample was prepared again if the extraction efficiency of MNV-1 was less than 1% [35]. The limit of detection (LOD) was calculated for the original volume of filtered water (LOD = (volume of final concentrated retentate × volume of eluate from nucleic acid extraction × volume of nucleic acid template added to the PCR reaction/volume of recovery efficiency from filtration × volume of original water samples)) [34].

### 2.4. Male-Specific Coliphage Isolation, Aerobic Plate Count (APC), and Coliform Count

Male-specific coliphages were isolated using the US Environmental Protection Agency method 1601 [42]. The enrichment of male-specific coliphages (obtained from the reservoir water samples) was conducted by mixing 100 mL of each sample with 1.25 mL of stock magnesium chloride (MgCl_2_), 0.5 mL of log-phase *Escherichia coli* Famp (ATCC 700891), 5 mL of 10 X tryptone soya broth (TSB), and 1 mL of stock ampicillin/streptomycin. The broth was incubated overnight at 37 °C. Subsequently, 10 μL of the enriched broth was spotted onto a premade spot plate coated with log phase *E. coli* Famp. The plate was observed after overnight incubation at 37 °C for a circular zone of lysis attributed to male-specific coliphages.

APC and coliform counting were performed using Petrifilm aerobic count plates and coliform count plates (3M, SD, USA), respectively [43,44]. Briefly, 1% peptone water was prepared in advance for dilution of water samples. Once the water sample arrived in the lab, 1 mL of each of the raw and diluted water samples was inoculated on the plate. Subsequently, the inoculated plates were incubated overnight at 37 °C. Thereafter, the colonies were counted after 24 h of incubation. On Petrifilm plates, red colonies with gas bubbles were counted as coliforms and blue colonies with gas bubbles as *E. coli*.

### 2.5. Weather and Environmental Data

The geographic information for each reservoir was obtained from the Korea rural community corporation [2]. Details regarding catchment area, pondage, and numbers of bungalow fishing spots, fishing sites, and animal farms near the reservoirs are summarized in Table 2. Data for daily average temperature, humidity, and precipitation were collected from the Korea meteorological agency [5]. Subsequently, the relationship between the accumulative precipitation and contamination by foodborne viruses and indicators microorganisms was determined [45,46]. In this regard, 3-day, 7-day, 10-day, and 14-day accumulative precipitation values were calculated retrogradely from the daily precipitation data.

### 2.6. Statistical Analyses

The Pearson correlation coefficient was estimated for analyzing the relationship between viruses and male-specific coliphages. Student’s T-test was performed for analyzing differences in the viral concentrations of the river and reservoir samples and the detection rates in the different precipitation levels. The correlation between weather/environmental factors and prevalence of viruses was analyzed through linear regression analysis (*Y* = *β*0 + *β*1*X*, where *Y* is the value of the dependent variable; *β*0 and *β*1 are the regression coefficients; and X is a known constant of independent variables). Subsequently, the data were log transformed. The analyses were conducted using SPSS Statistics (version 25.0) and statistical significance was defined as *p* < 0.05.

## 3. Results

### 3.1. Detection of Viruses and Indicator Microorganisms

The prevalence of viruses and indicator microorganisms in the reservoir and river samples is summarized in Table 3. Twenty-nine (30.2%) and 23 (23.9%) of the 96 irrigation water samples tested positive for viruses and male-specific coliphages, respectively. Among the six viruses, HuNoV GII indicated the highest detection rate (27.1%). The detection rates for AdenoV, RotaV, HuNoV GI, and AstroV were 10.4, 10.4, 4.2, and 3.1%, respectively. HAV was not detected in any sample.

The average populations of APC and coliforms in the reservoir samples ranged from 2.07 ± 0.82 to 3.33 ± 0.50 log CFU/mL and 1.15 ± 0.75 to 1.95 ± 0.80 log CFU/mL, respectively. *E. coli* was detected in four reservoir samples. The contamination of reservoir samples by *E. coli* was low (1~10 CFU/mL). The average populations of APC and coliforms in the river samples were 3.28 ± 0.29 and 2.10 ± 0.60 log CFU/mL, respectively. *E. coli* was detected in five of the six water samples (site K) and indicated a population of 7.2 ± 7.98 CFU/mL.

Geographically, site H was the most contaminated (Table 3). The detection rate of virus-positive water samples for sites E, F, G, I, and J were 33.3%. The water sample was twice positive for virus in sites A and C, and only once positive in site D. No viruses nor male-specific coliphages were detected in site B. The Anseong river (site K) was consistently positive for virus and male-specific coliphages except one time (Table 3).

In this study, the viral recovery efficiency of spiked MNV-1 showed 36.86% ± 27.96% (mean ± SD). This is regarded as the recovery efficiency of viruses. The detected viral loads range from −1.48 to 1.55 log_10_ genome copies/l (Figure 2). The average viral loads of HuNoV GI, HuNoV GII, AstroV, AdenoV, and RotaV in the reservoir samples were 0.70, −0.08, −0.23, −0.92, and 0.11 log10 genome copies/L, respectively, while those in the river samples were 0.74, 0.63, 0.08, 0.04, and 0.34 log_10_ genome copies/L, respectively. The reservoir and river samples indicated significantly different viral loads of HuNoV GII and AdenoV (*p* < 0.05). However, no significant difference was observed in the viral loads of HuNoV GI, AstroV, and RotaV between the reservoir and river samples.

The virus profiles for all samples are presented in Table 4. A total of 29 water samples were positive for 53 viruses. Fourteen of 29 water samples were positive only for a single virus. However, other water samples were contaminated with more than two viruses. Among the virus profile, HuNoV GII was the most common virus contaminating the reservoirs. The detection of RotaV or AdenoV along with HuNoV GII were frequently found in dual or multiple virus profiles. AstroV was infrequently detected only in multiple virus profiles. Although the isolation of male-specific coliphages was not matched with the isolation of *E. coli* in water samples, 18 (78.26%) of the 23 coliphage-positive water samples were associated with the detection of one or more viruses (*p* < 0.01) (Table 3 and Table 4). The Pearson correlation analysis indicated a significant correlation between the detection of HuNoV GII and male-specific coliphages (*p* < 0.01, Appendix A).

### 3.2. Seasonal Detection of Viruses and Indicator Microorganisms in Irrigation Water

Figure 3 presents the seasonal prevalence of viruses and male-specific coliphages in the reservoir and river samples (Figure 3). The seasonal detection rates of viruses in the 76 virus- or coliphage-positive samples were 40.78 and 39.47% during spring and autumn, respectively. The detection rate during summer was low (19.74%).

HuNoV GII was frequently detected during spring (46.15%, 12/26) and autumn (34.61%, 9/26) in contrast to summer (19.24%, 5/26). AdenoV was detected twice during autumn and four times each during spring and summer. RotaV was frequently detected during autumn (60%, 6/10) in contrast to spring or summer. Astrovirus did not indicate a specific seasonal pattern and was intermittently detected (1–2 times) during each season. Furthermore, 3, 9, and 11 samples collected in summer, spring, and autumn, respectively, were tested positive for coliphages.

### 3.3. Correlation between Detected Viruses and Weather Factors

Table 5 summarizes the correlations between detected viruses and weather factors. The results indicate that precipitation was significantly correlated with the prevalence of foodborne viruses, male-specific coliphages, HuNoV GII, and AdenoV (*p* < 0.05). However, humidity and temperature were not correlated with the prevalence of foodborne viruses and male-specific coliphages (*p* > 0.05). 

Subsequently, the relationship between contamination by foodborne viruses and accumulative precipitation was analyzed (Figure 4). The results indicate that the 10-day accumulative precipitation was closely related to virus detection in contrast to 7-day and 14-day accumulative precipitation values (data not shown). The detection rates of foodborne viruses for 10-day accumulative precipitation values ranging from 20–60, <20, and >60 mm were 71.7 (38/53), 20.75 (11/53), and 7.54% (4/53), respectively (Figure 4B). Corresponding to the detection rates, 10-day accumulative precipitation values greater than 60 mm indicated significantly lower viral loads in contrast to those less than 60 mm (*p* < 0.01) (Figure 4A).

### 3.4. Correlation between Detected Viruses and Environmental Factors

The relationships between detected viruses and environmental factors are presented in Table 5. Environmental factors, such as catchment area, pondage, and presence of livestock farms were not significantly related to the detection of viruses and male-specific coliphages (*p* > 0.05). The number of fishing sites was significantly correlated with the detection of HuNoV GII in the reservoirs (*p* < 0.05). Furthermore, the number of bungalow fishing spots was significantly correlated with the detection of male-specific coliphages (*p* < 0.05). As indicated in Table 2 and Table 3, foodborne viruses and male-specific coliphages were frequently detected in samples from sites C, F, H, I, and J. These sites were characterized by the presence of fishing sites and several bungalow fishing spots. Unlike regular fishing sites, bungalow fishing spots provided a unique facility for recreational fishing. Specifically, these spots were equipped with toilets and located above the reservoirs. Site B was a small reservoir without any fishing sites or bungalow fishing spots which foodborne viruses and male-specific coliphages were not detected during the sampling period. Site K (the river) was characterized by the downstream merging of sites C and D. The detection rate for male-specific coliphages and viruses were 21.73 (5/23) and 28.30% (15/53), respectively.

## 4. Discussion

To the best of our knowledge, this is the first field study to assess the prevalence of foodborne viruses and indicator microorganisms in reservoirs in Gyeonggi province, South Korea. Moreover, several previous studies have examined the effects of inland runoff and rainfall on the water quality of rivers, lakes, and underground water [45,47,48,49,50]. However, few studies have focused on the surface waters of reservoirs.

Several GCMs have predicted that drought hazard in the eastern and central regions of Korea contributed to the lasting severe drought in Gyeonggi province in 2017 [3]. Therefore, reservoirs in Gyeonggi province played an important role in supplying irrigation water to crops and fresh produce. In this study, indicative bacteria, coliphages, and enteric viruses were monitored for a period of 18 months because temporally variable weather factors, such as heavy rains, floods, and droughts can alter water quality [8,9].

### 4.1. Prevalence of Foodborne Viruses and Indicator Microorganisms in Reservoirs

HuNoV GII was the most frequently detected virus (27.1%) in the reservoir samples. Shin et al. reported that two (14.3%) of the 14 irrigation water samples tested positive for both HuNoV GI and GII [51]. Furthermore, their results indicated sporadic contamination of lettuce and strawberry by HuNoV or HAV in Korea [51]. Similarly, Cheong et al. detected HuNoV in one of the 29 groundwater samples collected from Gyeonggi province [47]. Previous studies have reported significant spatial variation in the detection rates of foodborne viruses in irrigation water. European countries, including Finland, the Czech Republic, Serbia, Poland, Italy, and Germany, have indicated low detection rates of HuNoV (3.6~8.3%) [17,52,53]. Contrarily, Argentina, Ghana, and Egypt have reported high detection rates of HuNoV in irrigation water (31~80%) [54,55,56]. Therefore, viral contamination of irrigation water sources (e.g., groundwater, wastewater, rivers, and reservoirs) should be monitored and controlled for ensuring the production of safe produce.

AdenoV is a promising viral indicator of surface water quality [57]. Verani et al. reported that the detection frequencies of AdenoV in various aquatic environments were higher than those of HuNoV [57]. This study concluded that HuNoV+AdenoV (and their combinations with other viruses) characterized the commonly detected viral profiles of contaminated reservoirs (Table 4). Although AdenoV and RotaV were less prevalent than HuNoV, the detection rate of AdenoV alone was as low as 10.4% in the sampled reservoirs. Similarly, a previous study reported that AdenoV was detected in four (13.8%) of the 29 irrigation water samples collected in Korea while RotaV was not detected [47]. In European countries, the detection rate of AdenoV was 28.1% in irrigation waters while that of RotaV in Italy was 1.4% [17,52]. However, it should be noted that the detection rate of AdenoV and RotaV were as high as 93.7 and 50%, respectively in irrigation waters in a developing country [55]. As the detection frequency of AdenoV was affected by the quality of irrigation water in each country and region, it could be an indicator to monitor the water quality and virus contamination.

In this study, HAV was not detected in the irrigation water samples; however, groundwater contamination has been linked to several HAV outbreaks in South Korea [28,29]. It should be noted that HAV can survive in groundwater and rivers for several months [29,58]. In addition, hepatitis E virus is a waterborne, foodborne, and also zoonotic virus with a burden increasing recently [23]. Therefore, the surveillance and control of HAV and HEV contamination in reservoirs and other water resources in the later studies are important to prevent waterborne or foodborne viral outbreaks in South Korea.

A previous study reported that the average viral loads of HuNoV was 2.30 log_10_ genome copies/L in a river in the Netherlands [59]. Similarly, the viral load of AdenoV ranged from 2.79 to 2.93 log_10_ genome copies/L in a river in Taiwan [60]. Furthermore, RotaV concentrations were as high as 4.14 log_10_ genome copies/L in surface river water in Germany [53]. However, in this study, the viral loads of viruses were low (−1.48 to 1.55 log_10_ genome copies/L). These results suggest that the reservoirs and river examined in this study indicated significantly lower viral contamination than water sources tested in previous studies. Therefore, an assessment of previous studies highlights the wide variation in the viral loads of different water samples. This discrepancy could be attributed to differences in water sample volumes and/or recovery and calculation methods utilized [30,45,61]. Although existing data on viral contamination of reservoirs are limited [62], viral loads in reservoirs in Korea are generally low and similar to those in underground water. Therefore, future studies should focus on nucleic acid sequencing of different viruses to allow identification and tracking of viral contaminants in reservoirs.

### 4.2. Seasonal Prevalence of Foodborne Viruses in Reservoirs

It is important that the detection rates of foodborne viruses in irrigation water were higher in spring and autumn than in summer because crops and other produce all require high quantities of irrigation water during these seasons. Notably, HuNoV GII was frequently detected during spring (46.15%, 12/26) and autumn (34.61%, 9/26). AdenoV was also detected during spring and summer (80%, 8/10) while RotaV was primarily detected during autumn (60%, 6/10). As 70% of the freshwater used for irrigation purposes is frequently contaminated with viruses worldwide, the risk of virus contamination in fresh produce is high during spring, summer, and autumn [18]. Furthermore, the HuNoV was detected in as much as 18.2~20.7% in irrigation water from a canal in Arizona, USA [63]. Notably, foodborne viral outbreaks associated with fresh produce such as lettuce had been reported in the last decades [64]. Therefore, contaminated irrigation water is a potential hazard to agricultural products during the growing and pre-harvest periods. Especially, virus contamination of irrigation water or reservoirs should be prevented during the pre-harvest period for the production of safe fresh produce.

In the aspect of public health, this study has some limitations to not examine the virus contamination in winter because the reservoirs were frozen. HuNoVs and RotaV frequently cause outbreaks in winter because the cold weather protects the virus infectivity [65]. Some studies highlighted the non-seasonal detection of foodborne viruses in aqueous environments [10,13]. Regardless of the season, AdenoV and JC polyomavirus were frequently detected in the wastewaters and surface waters of the United Kingdom [66]. Although a significant seasonal pattern was not found in the prevalence of foodborne viruses, virus contamination may not be neglected because a lot of water is used for fresh produce in the green house during the winter. Therefore, the viral load and viability of irrigation water and reservoirs should be investigated in a further study.

### 4.3. Effects of Weather Factors on the Prevalence of Foodborne Viruses in Reservoirs

The results of this study indicate that rainfall was significantly correlated with the prevalence of foodborne viruses (HuNoV GII and AdenoV) and male-specific coliphages in the reservoirs (Table 5). Previous studies have also identified rainfall as a critical factor affecting the concentrations of pathogenic microorganisms in irrigation water and seawater [67,68]. In particular, rainfall can alter the concentrations of indicative microorganisms in surface waters through runoff and resuspension of bottom sediments [69]. As indicated in a previous study, heavy rainfall increased E. coli contamination levels in California [70]. Similarly, 70 mm of rainfall decreased E. coli concentrations (<100 CFU/100 mL) in Arizona while no precipitation increased E. coli concentrations (400 CFU/100 mL) in the Rio Grande water [71]. The effect of rainfall on microbial contamination of surface waters has been shown to vary by region, the number of consecutive rainfall events, and flush levels [70].

In this study, the water samples in reservoirs indicated a high detection frequency of foodborne viruses (71.69%, 38/53) for accumulative precipitation levels ranging from 20 to 60 mm. On the contrary, detection rates and viral loads were low for accumulative precipitation >60 mm. Therefore, moderate-level rainfall events likely increased runoff and resuspension of sediments, thus resulting in the viral contamination of reservoirs. Korea experiences monsoon rains from June to July and intermittent typhoon rains in August [72]. Consecutive moderate-level rainfall events during monsoons may increase the viral contamination of irrigation waters. Contrarily, heavy rains (>60 mm) during typhoons may decrease the contamination of irrigation waters. Furthermore, as indicated in Figure 2, the viral loads of HuNoV GII and AdenoV were significantly higher in the river samples than in the reservoir samples. However, the average viral loads of HuNoV GI, AstroV, and RotaV were not significantly different between the river and reservoir samples. As the correlation analysis presented that the prevalence of HuNoV GII and AdenoV were significantly affected by the rainfall, we concluded that HuNoV and AdenoV from the sediment in the river were easy to be resuspended because of the rainfall. A previous report indicated that the lack of rainfall or droughts can concentrate surface waters and increase microbial concentration levels [10]. The Anseong city experienced a severe drought from May to July 2017. Consequently, the level of irrigation water in the reservoirs was extremely low. Subsequently, algal blooms were observed which further deteriorated the water quality. Moreover, the irrigation water was highly concentrated, and viruses were not detected until the commencement of rainfall in August 2017. Therefore, the concentration of irrigation water during the drought was likely unrelated to virus detection in the reservoirs in this study.

Depending on the conditions, foodborne viruses can survive in different environments for several months [11,12]. Several weather factors, including rainfall, drought, humidity, temperature, and sunlight, affect the microbial infectivity [12]. Especially, low temperature, 10–66% relative humidity, and rainfall are closely associated with HuNoV outbreaks [73]. As Korean spring and autumn has a low temperature, low relative humidity, and moderate rainfall, such an environmental factor may explain that HuNoV was frequently detected in these seasons rather than in summer.

### 4.4. Effects of Environmental Factors on the Prevalence of Foodborne Viruses in Reservoirs

This study analyzed the effects of environmental factors (i.e., catchment area, pondage, and numbers of livestock farms, bungalow fishing spots, and fishing sites near the reservoirs) on the prevalence of viruses and indicator microorganisms in the reservoirs. Previous studies have indicated that irrigation water is susceptible to contamination by fecal waste from wild animals, livestock, and humans [15,18,74,75]. Specifically, the numbers of fishing sites and bungalow fishing spots were significantly correlated with the prevalence of HuNoV GII and male-specific coliphages, respectively. Notably, sites C, H, I, and J were characterized by several bungalow fishing spots which were equipped with toilets. Therefore, leakage of human excreta likely directly contaminated the reservoir water. Consequently, the detection frequencies of viruses and indicator male-specific coliphages were higher for sites C, H, I, and J in contrast to the other sites. Additionally, dual or multiple viruses were detected frequently at sites C, H, I, and J. The presence of toilets near irrigation water sources has been identified as a risk factor for virus transmission [76,77]. Therefore, potential contamination sources, such as bungalow fishing spots and fishing sites should be eliminated or minimized in reservoir areas. Additionally, facilities near reservoirs should ensure optimal management of septic systems to prevent potential viral contamination [18]. Site B was characterized by a small reservoir without fishing sites or bungalow fishing spots. Notably, viruses, E. coli, and male-specific coliphages were not detected at this site.

The catchment area and the number of livestock farms near the reservoirs were not significantly correlated with the detection of viruses. Contrarily, the detection of hepatitis E virus in streams in Iowa, USA was correlated with runoff and livestock manure [78]. Guber et al. developed a model using the soil and water assessment tool of ARC-GIS to predict the transmission of bacterial pathogens between livestock and deer in the USA. They analyzed the distribution of pathogens in a watershed [74]. For enhancing the management of surface waters and groundwater a geographic information system–multicriteria decision analysis-based model was adopted to assess the quality of irrigation water [79,80]. However, these models are not optimized for analyzing viral contamination in reservoirs. Therefore, future studies should focus on developing better predictive models for elucidating viral distributions in reservoirs and determining their correlations with various weather and environmental factors.

## 5. Conclusions

This study indicated that the environmental and weather factors were closely associated with the prevalence of foodborne viruses in reservoirs in South Korea. Especially, precipitation was highly correlated with the virus detection and male-specific coliphages. Furthermore, fishing sites and recreational facilities were identified as important sources of viral contamination in the reservoirs. As reservoirs play an important role in providing irrigation water, the microbiological monitoring and elimination of a contamination source will be needed to control the risk of viral contamination.

## Figures and Tables

**Figure 1 microorganisms-08-01224-f001:**
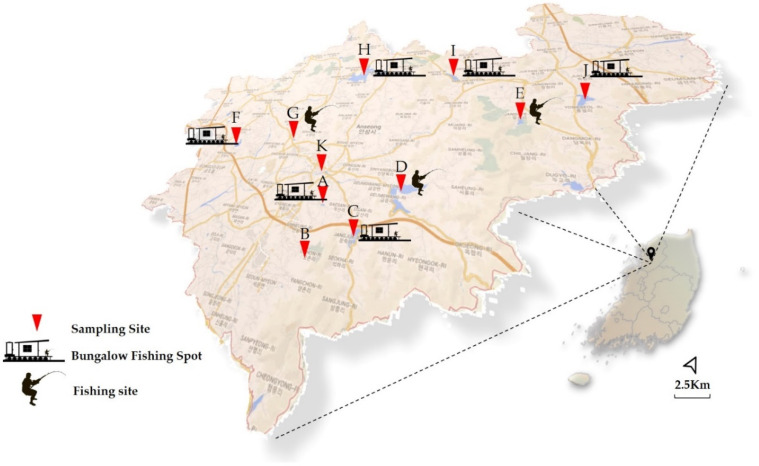
Geographical locations of ten reservoirs and Anseong River (site K) in Gyeonggi province, South Korea. Site K is characterized by the downstream merging of sites C and D. Fishing sites and bungalow fishing spots represent recreational facilities.

**Figure 2 microorganisms-08-01224-f002:**
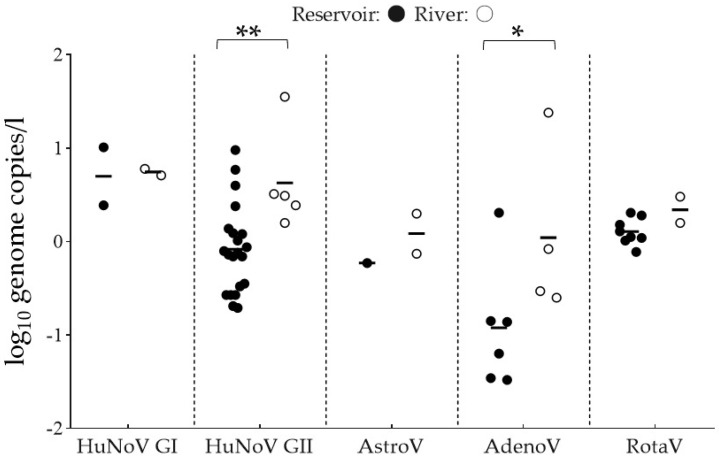
Average viral loads of foodborne viruses detected in the reservoirs and Anseong River. ● and ○ indicate reservoir and river samples, respectively. – means average viral loads. ** *p* < 0.01; * *p* < 0.05.

**Figure 3 microorganisms-08-01224-f003:**
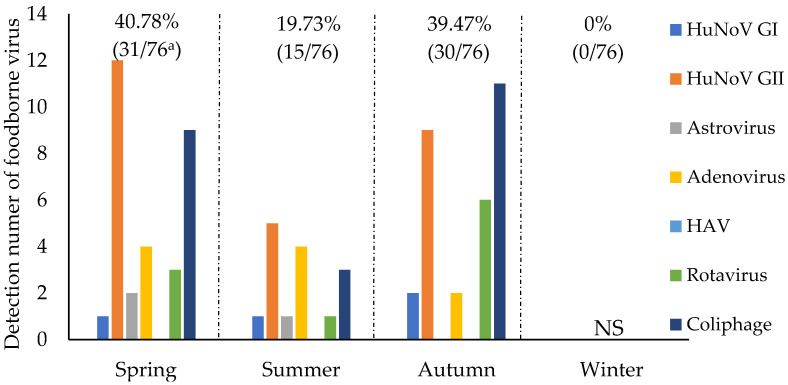
Seasonal prevalence of viruses and male-specific coliphages from May 2017 to November 2018; ^a^ numerator represents a number of detected viruses and male-specific coliphages; denominator represents a number of all viruses and male-specific coliphages. Spring (March–May); summer (June–August); autumn (September–November); winter (December–February).

**Figure 4 microorganisms-08-01224-f004:**
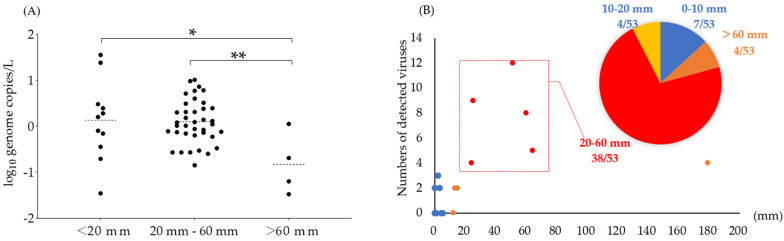
Relationship between precipitation and virus detection. (**A**) Correlation between viral loads (*x*-axis) and accumulative precipitation (*y*-axis). (**B**) Detection rates of viruses in response to 10-day accumulative precipitation.

**Table 1 microorganisms-08-01224-t001:** Primers and probes utilized for the seven viruses examined in this study.

Virus	Primer/Probe	Oligonucleotide Sequence	Amplification Conditions	Reference
HuNoV GI	QNIF4	5′-CGCTGGATGCGNTTCCAT-3′	95 °C 15 s, 60 °C 1 min (45 cycles)	[36]
	NV1LCR	5′-CCTTAGACGCCATCATCATTTAC-3′		
	NVGG1p	5′-FAM-TGGACAGGAGAYCGCRATCT–TAMRA-3′		
HuNoV GII	QNIF2	5′-ATGTTCAGRTGGATGAGRTTCTCWGA-3′	95 °C 15 s, 60 °C 1 min (45 cycles)	[36]
	COG2R	5′-TCGACGCCATCTTCATTCACA-3′		
	QNIFs	5′-FAM-AGCACGTGGGAGGGCGATCG-TAMRA-3′		
HAV	Forward P	5′-GGTAGGCTACGGGTGAAAC-3′	95 °C 10 s, 55 °C 20 s (45 cycles)	[37]
	Reverse P	5′-AACAACTCACCAATATCCGC-3′		
	HAV Probe	5′-FAM-CTTAGGCTAATACTTCTATGAAGAGATGC-TAMRA-3′		
RotaV	MVP3-FDeg	5′-ACCATCTWCACRTRACCCTC-3′	95 °C 15 s, 56 °C 1 min (45 cycles)	[38]
	MVP3-R1	5′-GGTCACATAACGCCCCTATA-3′		
	MVP3-Probe	5′-FAM-ATGAGCACAATAGTTAAAAGCTAACACTGTCAA-TAMRA-3′		
AstroV	AstV F	5′-CCDGCCAGRCTCACAGAAGAG-3′	94 °C 15 s, 55 °C 20 s (45 cycles)	[39]
	AstV R	5′-GACTTGCTAGCCATCACACTYC-3′		
	AstV Probe	5′-FAM-ACTCCATCGCATTTGGAGGGGAGGACC-TAMRA-3′		
AdenoV	JTVFF	5′-AACTTTCTCTCTTAATAGACGCC-3′	95 °C 10 s, 55 °C 30 s, 72 °C 27 s (45 cycles)	[40]
	JTVFR	5′-AGGGGGCTAGAAAACAAAA-3′		
	JTVFAP	5′-FAM-CGAAGAGTGCCCGTGTCAGC-BHQ-3′		
MNV-1	MNV 5036	5′-ACGCTCAGCAGTCTTTGTGA-3′	95 °C 15 s, 60 °C 45 s (45 cycles)	[41]
	MNV 5088	5′-CTGGCCTCAGAGCCATTG-3′		
	MNV 5060	5′-FAM-CGCTGCGCCATCACTCATCC-TAMRA-3′		

HuNoV: Human Norovirus; HAV: Hepatitis A Virus; RotaV: Rotavirus; AstroV: Astrovirus; AdenoV: Adenovirus; MNV-1: Murine Norovirus.

**Table 2 microorganisms-08-01224-t002:** Environmental characteristics of the sampling sites.

Reservoir	Longitude/Latitude	Catchment Area (ha)	Pondage (m^3^)	No. of Livestock Farms	No. of Bungalow Fishing Spots	No. of Fishing Sites
Site A	36°59′03.0″ N 127°17′04.6″ E	121	45	6	0	1
Site B	36°57′45.0″ N 127°16′56.8″ E	60	3.5	0	0	0
Site C	36°58′14.4″ N 127°18′40.4″ E	1240	4707	10	12	1
Site D	36°59′45.0″ N 127°19′54.5″ E	4830	12,085	34	0	1
Site E	37°02′32.8″ N 127°24′26.3″ E	225	992.4	0	0	1
Site F	37°01′08.3″ N 127°11′08.0″ E	375	614	18	2	1
Site G	37°01′30.0″ N 127°14′28.1″ E	105	39	3	0	1
Site H	37°04′57.6″ N 127°16′59.7″ E	7100	15,217	170	105	4
Site I	37°04′36.0″ N 127°21′54.2″ E	485	1859	3	9	1
Site J	37°03′27.3″ N 127°26′43.3″ E	790	2932.4	7	14	2
Site K	37°00′08.0″ N 127°16′02.3″ E	172,200	-	-	-	-

**Table 3 microorganisms-08-01224-t003:** Prevalence of foodborne viruses and indicator microorganisms in the ten reservoirs and a river.

Sampling Site	HuNoV GI	HuNoV GII	AstroV	AdenoV	HAV	RotaV	Detection Rate of Positive Samples	Indicator Microorganisms			
								Male-Specific Coliphage	APC (log CFU/mL)	Coliform (log CFU/mL)	*E. coli* (CFU/mL)
A	0/9	2/9	0/9	1/9	0/9	0/9	2/9 (22.22%)	1/9	2.86 ± 0.97	1.15 ± 0.75	ND ^b^
B	0/9	0/9	0/9	0/9	0/9	0/9	0/9 (0.00%)	0/9	3.33 ± 0.50	1.81 ± 0.26	ND
C	1/9	2/9	0/9	2/9	0/9	1/9	2/9 (22.22%)	3/9	2.07 ± 0.82	1.44 ± 0.77	ND
D	0/9	1/9	0/9	0/9	0/9	0/9	1/9 (11.11%)	0/9	2.63 ± 0.77	1.49 ± 0.65	1 ± 0
E	0/9	2/9	0/9	1/9	0/9	0/9	3/9 (33.33%)	0/9	2.79 ± 0.67	1.95 ± 0.80	ND
F	0/9	2/9	0/9	1/9	0/9	0/9	3/9 (33.33%)	1/9	3.14 ± 0.61	1.53 ± 0.80	ND
G	1/9	3/9	0/9	0/9	0/9	2/9	3/9 (33.33%)	1/9	2.72 ± 0.60	1.84 ± 0.53	2.5 ± 2.12
H	0/9	3/9	0/9	0/9	0/9	2/9	4/9 (44.44%)	2/9	2.50 ± 0.65	1.60 ± 0.63	ND
I	0/9	3/9	1/9	1/9	0/9	2/9	3/9 (33.33%)	6/9	2.63 ± 0.51	1.69 ± 0.56	10 ± 0
J	0/9	3/9	0/9	0/9	0/9	1/9	3/9 (33.33%)	4/9	2.32 ± 0.27	1.27 ± 0.60	1 ± 0
K	2/6	5/6	2/6	4/6	0/6	2/6	5/6 (83.33%)	5/6	3.28 ± 0.29	2.10 ± 0.60	7.2 ± 7.98
Total	4/96 ^a^	26/96	3/96	10/96	0/96	10/96	29/96	23/96	-	-	-
	(4.16%)	(27.1%)	(3.12%)	(10.4%)	(0.00%)	(10.4%)	(30.2%)	(23.9%)			

^a^ A number of positive samples/total numbers of samples. ^b^ ND: Not detected.

**Table 4 microorganisms-08-01224-t004:** Profiles of viruses and male-specific coliphages detected in the reservoirs.

Detection of Viruses and Male-Specific Coliphages		No. of Positive Samples
Single	HuNoV GII	6
	HuNoV GII + Male-specific coliphage	5
	AdenoV	2
	RotaV	1
Double	HuNoV GII + RotaV	2
	HuNoV GII + RotaV + Male-specific coliphage	4
	HuNoV GII + AdenoV + Male-specific coliphage	2
Multiple	HuNoV GII + AdenoV + AstroV + Male-specific coliphage	2
	HuNoV GII + AdenoV + RotaV + Male-specific coliphage	1
	HuNoV GI + HuNoV GII + RotaV + Male-specific coliphage	1
	HuNoV GI + HuNoV GII + AdenoV + Male-specific coliphage	1
	HuNoV GI + HuNoV GII + AdenoV + AstroV + Male-specific coliphage	1
	HuNoV GI + HuNoV GII + AdenoV + RotaV + Male-specific coliphage	1
Total positive samples (positive rate)		29 (30.2%) ^a^

^a^ A total of 96 samples was collected for virus detection from the surface irrigation water of the reservoirs.

**Table 5 microorganisms-08-01224-t005:** Results of linear regression analysis between the number of positive samples for viruses/male-specific coliphages and weather/environmental factors.

	Log_10_*β* (SE *b*) ^a^							
	Virus	Male-Specific Coliphages	HuNoV GII	AdenoV	RotaV	AstroV	HuNoV GI	HAV
Weather factors								
Precipitation ^b^	0.714 (0.095) *	0.713 (0.030) *	0.706 (0.012) *	0.642 (0.053) *	0.511 (0.067)	0.237 (0.044)	0.452 (0.044)	ND ^c^
Humidity	0.366 (1.279)	0.413 (0.388)	0.428 (0.148)	0.335 (0.651)	0.072 (0.776)	−0.003 (0.449)	0.331 (0.466)	ND
Temperature	−0.025 (0.315)	−0.002 (0.098)	0.006 (0.038)	0.412 (0.144)	−0.409 (0.163)	0.248 (0.100)	−0.042 (0.113)	ND
Environmental factors								
Catchment area	0.233 (0.139)	0.263 (0.143)	0.296 (0.092)	−0.130 (0.094)	0.281 (0.110)	−0.009 (0.049)	−0.110 (0.065)	ND
Pondage	0.432 (0.076)	0.385 (0.081)	0.489 (0.050)	0.107 (0.055)	0.296 (0.065)	0.130 (0.028)	−0.090 (0.038)	ND
Livestock	0.267 (0.140)	0.027 (0.145)	0.356 (0.092)	−0.180 (0.095)	0.281 (0.112)	−0.157 (0.049)	−0.065 (0.066)	ND
Bungalow fishing spot	0.530 (0.117)	0.737 (0.097) *	0.307 (0.080)	0.002 (0.091)	0.663 (0.083)	0.207 (0.046)	−0.016 (0.630)	ND
Fishing site	0.605 (0.451)	0.480 (0.518)	0.745 (0.255) *	−0.159 (0.371)	0.535 (0.384)	−0.055 (0.193)	−0.083 (0.257)	ND

^a^ β: Mean value of the regression coefficient; SE *b*: Standard error of the regression coefficient. * *p* < 0.05. ^b^ Ten-day accumulative precipitation values and average values of humidity and temperature were collected during the sampling period. ^c^ HAV was not detected.

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
