# Peer review of "Effects of Weather and Environmental Factors on the Seasonal Prevalence of Foodborne Viruses in Irrigation Waters in Gyeonggi Province, Korea"

_microorganisms, 2020, doi:10.3390/microorganisms8081224_

Round 1

Reviewer 1 Report

The authors have studied virological quality of irrigation water in Korea. They have detected RNA of multiple human gastroenteritis viruses in the irrigation water samples. It means that water is contaminated by excreta or sewage of human origin. The manuscript is generally well written, but it is recommended that the English language is checked. Some data should be added for clarity, such as description of reservoirs and origin of water in the reservoirs. Below, specific comments (not a complete list) are given for further improvement of the manuscript.

Abstract: No mention on indicators? Add also, that viral RNA was detected (or method).

L23 Check ’reservoir’ or ’reservoirs’.

L25, L28 ’hepatitis A virus’ and ’norovirus’ without capital letters

L30 Check if ’was’ is necessary here.

L73 Consider mentioning hepatitis E virus as well.

L83 Check ’indicative microorganisms’. Usually ’indicator microorganisms’, or ’sanitary-indicative microorganisms’ is used. Check throughout the manuscript.

L89-90 More description of the reservoirs is required. Are they natural or man-made? Where is the water coming to them? Are there other rivers in the region? More data on Ankeong river is needed. Are there big cities located along it? Is this densely populated region? Are there wastewater treatment plants along the river and do they discharge treated wastewater into the river?

L103 or elsewhere: Please add the detection limit of the method. Can you give estimates of the recovery values? Were they taken into account in calculations of virus concentrations?

L110 So, was 0,5 liter the elution volume? Is this the elution method?

L114, L135 Check subscriptions for chemicals.

L116 Check ’mins’ or ’min’.

Figure 1. Is it possible to give more description of the fishing sites and bungalow spots? How frequently are they visited? Which kind of sanitary solutions do they have?

L124 How about reverse transcription? No mention in the text or table. Did the authors include a PCR-inhibitor control in the samples?

L141-4 References are missing. In addition, E. coli method is missing. Also, remember to write names of bacteria in italics. Check throughout the text.

L181 ’No viruses nor male-specific ...’

L182 coliphages

L183 Refer to table or figure.

L193 The sentence starting with ’A total of 53 viruses were detected...’ requires modification. In fact, this paragraph might be rewritten and the focus should be on the most important results (not repeating all data in the table).

Figure 2 legend. Indicate the lines (average values?)

Table 3. Modify the legend (’surface irrigation water’ is a bit clumsy). Footnote: Do the authors mean ’a number of positive samples/total number of samples’?

Table 4. Footnote. Please, modify the sentence (A total of 96 samples...)

L220 Furthermore, 3, 9 and 11 samples collected in summer, spring and autumn, respectively, were tested positive for coliphages.

Figure 3. Please, rewrite the legend. For example, ’a number of all viruses’ instead of ’whole detected numbers of viruses’. Check uppercases and lowercases.

L233 Check if it should be ’lower viral loads’ instead of ’low viral loads’.

L249 This is a bit odd sentence, since presence of these human (?) viruses in any water tells mostly about fecal contamination of human origin.

L252 Please, rewrite the sentence (not ’concentrations’).

Figure 4. Check the y-axis: I suppose it is not ’number of detected viruses’ but ’detection rate of viruses’?

Discussion: I think it is important to mention in the discussion that this study does not tell data on infectious viruses (only viable bacteriophages).

Table 5. Please, check the legend (rephrase ’the detection numbers of viruses’). In the table, is it possible to indicate a value above which the analysis is regarded as positive?

L273 Add the citation for Cheong et al, here, and L287 for Verani et al.,. Check throughout the manuscript.

L281-291 The paragraph could be rewritten. Try to draw some conclusions (write discussion) from those results and facts. Did results of your study agree with others? If not, what would be the reasons?

L284 Better: ’significantly lower...than...’. Check also L307. Please, correct.

L285 Do the authors really mean viral load (meaning viral concentration in water) or prevalence or detection rate?

L289 The sentence should reveal that it is question about some other study than this one.

L298 Avoid writing results again in the discussion. It is not necessary.

L311 check ’existing’

L313 ’nucleic acid sequencing’

L318 ’Furthermore’ instead of ’further’. Please, check throughout the manuscript.

L319-330 Please, include winter in the discussion of seasonality.

L323 To my understanding, the results of this study are not suggesting this statement to be true. Please, reconsider.

L355-6 Reference is required.

L358 Please, add ’in this study’ in the sentence.

L362 Discussion on viral infectivity might be added in this paragraph.

L70-4 Please, rewrite these sentences for clarity. Try to avoid repetitions.

L381 Yes, HEV is zoonotic virus.

L383 Add citation (reference) for Guber et al.

L385-388 These sentences should be rewritten for clarity.

L392 ’confirmed’ is quite strong expression. Please, modify.

Author Response

To Reviewer #1:
The authors have studied virological quality of irrigation water in Korea. They have detected RNA of multiple human gastroenteritis viruses in the irrigation water samples. It means that water is contaminated by excreta or sewage of human origin. The manuscript is generally well written, but it is recommended that the English language is checked. Some data should be added for clarity, such as description of reservoirs and origin of water in the reservoirs. Below, specific comments (not a complete list) are given for further improvement of the manuscript.

1) L23 Check ’reservoir’ or ’reservoirs’. 

  • We changed ’reservoir’ into ’reservoirs’ in L23.

2) L25, L28 ’hepatitis A virus’ and ’norovirus’ without capital letters

  • They were corrected.

3) L30 Check if ’was’ is necessary here.

  • We deleted ‘was’ in L30.

4) L73 Consider mentioning hepatitis E virus as well.

  • We added the HEV in this sentence.

5) L83 Check ’indicative microorganisms’. Usually ’indicator microorganisms’, or ’sanitary-indicative microorganisms’ is used. Check throughout the manuscript.

  • As you commented, we changed ’indicative microorganisms’ into ’indicator microorganisms’ in the manuscript.

6) L89-90 More description of the reservoirs is required. Are they natural or man-made? Where is the water coming to them? Are there other rivers in the region? More data on Anseong river is needed. Are there big cities located along it? Is this densely populated region? Are there wastewater treatment plants along the river and do they discharge treated wastewater into the river?

  • We respect your comments. Anseong is rural area with many agricultural farms. As many reservoirs have been constructed to supply irrigation water, they are located near farm area rather than hometown. As these reservoirs can be utilized for recreation purpose like fishing and other activity, we describe the characteristics of reservoirs as follows.
  • “Anseong is the city surrounded with many agricultural farms. The population of Anseng city is 184,287. This study selected ten man-made reservoirs to supply irrigation water to the Anseong region, Gyeonggi province of South Korea (Figure 1). The Anseong river (site K; characterized by the downstream merging of sites C and D) was also sampled. Site A, site F, site G are small reservoirs located in a flat area surrounded by arable farms and villages. Site D is the large reservoir with a broad catchment area, poundage, and high forest coverage. Site B and site E are located near a golf course with high forest coverage. Many bungalow fishing spots and recreational facility were present in Site C, H, I, and J.”
  • Although you mentioned other river and wastewater treatment plants, that is not directly related with the focus of this study. We paid attention on the virus contamination in reservoirs. As anseong river is located in the downstream of site C and site D, site K was additionally chosen to monitor and trace the virus contamination. We hope that these answer your questions properly.

7) L103 or elsewhere: Please add the detection limit of the method. Can you give estimates of the recovery values? Were they taken into account in calculations of virus concentrations?

  • Calculation method of LOD was added in L 149-152. We also added the calculation method of recovery in L 147-148, recovery rate which was presented in result L 216. (36.86% ± 27.96% (mean ±SD))

8) L110 So, was 0,5 liter the elution volume? Is this the elution method?

  • Yes, 0.5 liter is the elution volume. A volume of 0.5-liter Beef extract was used to elute the virus in the 1-MDS filter. This method is based on virus adsorption-elution (VIR-AD-EL) technique for the previous study (Lee H, Kim M, Lee JE, Lim M, Kim M, Kim JM, Jheong WH, Kim J, Ko G. May 1 2011. Investigation of norovirus occurrence in groundwater in metropolitan Seoul, Korea. Science of the Total Environment, 409 (11), 2078-84. doi:10.1016/j.scitotenv.2011.01.059).

9) L114, L135 Check subscriptions for chemicals.

  • The subscription of the chemicals was revised.

10) L116 Check ’mins’ or ’min’.

  • We corrected them.

11) Figure 1. Is it possible to give more description of the fishing sites and bungalow spots? How frequently are they visited? Which kind of sanitary solutions do they have?.

  • We respect these comments. As the reviewer#1 pointed, we provided the supplementary figures and graphical abstract with revised manuscript. As these facilities were constructed for the recreation, the number of people is changing depending on the season.

12) L124 How about reverse transcription? No mention in the text or table. Did the authors include a PCR-inhibitor control in the samples?

  • Thank you for your critical review. For all the amplicon of viruses, one step PCR was utilized for a cDNA (complementary DNA) synthesis and then amplification of the DNA template. We rewrote the details about reagents of the amplification as follows (L141-L143)
  • “(Virus RNA was amplified in a 25 µl reaction mixture containing of 5 µl of each sample RNA, 12.5 µl of 2x QuantiTect Probe RT-PCR Master Mix (QIAGEN, Germany), 0.25 µl of QuantiTect RT Mix, 500 nM of forward and reverse primer, 250 nM of probe)”

13) L141-4 References are missing. In addition, E. coli method is missing. Also, remember to write names of bacteria in italics. Check throughout the text.

  • Thank you for your recommendation, we revised this paragraph thoroughly.

14) L181 ’No viruses nor male-specific ...’

  • Thank you. ‘and’ was changed into ‘nor’.

15) L182 coliphages.

  • Corrected.

16) L183 Refer to table or figure.

  • They are revised.

17) L193 The sentence starting with ’A total of 53 viruses were detected...’ requires modification. In fact, this paragraph might be rewritten and the focus should be on the most important results (not repeating all data in the table).

  • We agree with the reviewer, therefore, we rewrote the sentence as follows.
  • “The virus profiles for all samples are presented in Table 4. A total of 29 water samples were positive for 53 viruses. Fourteen of 29 water samples were positive only for single virus. However, other water samples were contaminated with more than two viruses. Among the virus profile, HuNoV GII was the most common virus contaminating the reservoirs. The detection of RotaV or AdenoV along with HuNoV GII were frequently found in dual or multiple virus profile. AstroV was infrequently detected only in multiple virus profile. Although the isolation of male-specific coliphages was not matched with the isolation of E. coli in water samples, 18 (78.26%) of the 23 coliphage-positive water samples were associated with the detection of one or more viruses (P < 0.01) (Table 3 & 4). The Pearson correlation analysis indicated a significant correlation between the detection of HuNoV GII and male-specific coliphages (P < 0.01, Supplementary Figure 1).”

18) Figure 2 legend. Indicate the lines (average values?)

  • Yes, it is. We revised Figure 2 Legend.

19) Table 3. Modify the legend (’surface irrigation water’ is a bit clumsy). Footnote: Do the authors mean ’a number of positive samples/total number of samples’?

  • We revised the footnote and deleted the ‘surface irrigation water’.

20) Table 4. Footnote. Please, modify the sentence (A total of 96 samples...).

  • Thank you for your recommendation. We revised the footnote accordingly.

21) L220 Furthermore, 3, 9 and 11 samples collected in summer, spring and autumn, respectively, were tested positive for coliphages.

  • It is revised by following your comment in L249.

22) Figure 3. Please, rewrite the legend. For example, ’a number of all viruses’ instead of ’whole detected numbers of viruses’. Check uppercases and lowercases.

  • Thank you for your recommendation. Additional legend of Figure 3 was rewrote accordingly.

23) L233 Check if it should be ’lower viral loads’ instead of ’low viral loads’..

  • Yes. Thank you for your recommendation. We agree with reviewer. Therefore ’low viral loads’ was revised to ’lower viral loads’

24) L249 This is a bit odd sentence, since presence of these human (?) viruses in any water tells mostly about fecal contamination of human origin.

  • Thank you for your critical review. This sentence was rewritten in L277-279.

25) L252 Please, rewrite the sentence (not ’concentrations’).

  • This sentence was revised.

26) Figure 4. Check the y-axis: I suppose it is not ’number of detected viruses’ but ’detection rate of viruses’?

  • Thank you for your recommendation. But Y-axis in Figure 4B presents the number of viruses detected in water. Figure 4B showed the relationship of accumulative precipitation and virus detection. Based on the accumulative precipitation, the different colored dot (red, blue, yellow, orange) in the axis was marked with the number of detected virus. The pie chart presented that 20-60 mm precipitation was closely related with virus detection.

27) Discussion: I think it is important to mention in the discussion that this study does not tell data on infectious viruses (only viable bacteriophages).

  • We revised the discussion section thoroughly. It was shown in L 396-401.

28) Table 5. Please, check the legend (rephrase ’the detection numbers of viruses’). In the table, is it possible to indicate a value above which the analysis is regarded as positive?

  • Thank you for your critical review. Accordingly, ’the detection numbers of viruses’ was revised to ‘the number of positive samples for’. From the value numerical value, though it presented significantly when it >6, the explicit value and its model need further evaluation.

29) L273 Add the citation for Cheong et al, here, and L287 for Verani et al. Check throughout the manuscript.

  • Thank you for your recommendation. Accordingly, the reference of Cheong et al and Verani et al. were cited in L302 and L311.

30) L281-291 The paragraph could be rewritten. Try to draw some conclusions (write discussion) from those results and facts. Did results of your study agree with others? If not, what would be the reasons?

  • We rewrote this paragraph thoroughly as follows.
  • “AdenoV is a promising viral indicator of surface water quality [73]. Verani et al. reported that the detection frequencies of AdenoV in various aquatic environments were higher than those of HuNoV [73]. This study concluded that HuNoV+AdenoV (and their combinations with other viruses) characterized the commonly detected viral profiles of contaminated reservoirs (Table 4). Although AdenoV and RotaV were less prevalent than HuNoV, the detection rate of AdenoV alone was as low as 10.4% in the sampled reservoirs. Similarly, previous study reported that AdenoV was detected in four (13.8%) of the 29 irrigation water samples collected in Korea while RotaV was not detected [3]. In European countries, the detection rate of AdenoV was 28.1% in irrigation waters while that of RotaV in Italy was 1.4% [7, 36]. However, it should be noted that the detection rate of AdenoV and RotaV were as high as 93.7 and 50%, respectively in irrigation waters in developing country [64]. As the detection frequency of AdenoV was affected by the quality of irrigation water in each countries and regions, it could be an indicator to monitor the water quality and virus contamination.”

31) L284 Better: ’significantly lower...than...’. Check also L307. Please, correct.

  • Thank you for this point. This paragraph was re-written thoroughly.

32) L285 Do the authors really mean viral load (meaning viral concentration in water) or prevalence or detection rate?

  • Thank you for this point. This paragraph was re-written thoroughly.

33) L289 The sentence should reveal that it is question about some other study than this one.

  • We rewrote this paragraph accordingly.

34) L298 Avoid writing results again in the discussion. It is not necessary.

  • We rewrote it from L309-320

35) L311 check ’existing’.

  • Yes. it was revised.

36) L313 ’nucleic acid sequencing’.

  • Yes, ‘nucleic’ was added in the sentence for clarity.

37) L318 ’Furthermore’ instead of ’further’. Please, check throughout the manuscript.

  • Thank you for your recommendation. ‘Further’ was revised to ‘Furthermore’ in whole manuscript.

38) L319-330 Please, include winter in the discussion of seasonality.

  • We revised the section 4.2 thoroughly.

39) L323 To my understanding, the results of this study are not suggesting this statement to be true. Please, reconsider...

  • We revised the section 4.2 as follows.
  • “It is important that the detection rates of foodborne viruses in irrigation water were higher in spring and autumn than in summer because crops and other produce all require high quantities of irrigation water during these seasons. Notably, HuNoV GII was frequently detected during spring (46.15%, 12/26) and autumn (34.61%, 9/26). AdenoV was also detected during spring and summer (80%, 8/10) while RotaV was primarily detected during autumn (60%, 6/10). As 70% of the freshwater used for irrigation purposes is frequently contaminated with viruses worldwide, the risk of virus contamination in fresh produce is high during spring, summer, and autumn [76]. Furthermore, the HuNoV was detected as much as 18.2~20.7% in irrigation water from a canal in Arizona, USA [28]. Notably, foodborne viral outbreaks associated with fresh produce such as lettuce had been reported in the last decades [63]. Therefore, contaminated irrigation water is a potential hazard to agricultural products during the growing and pre-harvest periods. Especially, virus contamination of irrigation water or reservoirs should be prevented during the pre-harvest period for the production of safe fresh produce.
  • In the aspect of public health, this study has some limitation not to examine the virus contamination in winter because the reservoirs were frozen. HuNoVs and RotaV frequently cause outbreaks in winter because the cold weather protects the virus infectivity [11]. Some studies highlighted the non-seasonal detection of foodborne viruses in aqueous environments [52, 68]. Regardless of the season, AdenoV and JC polyomavirus were frequently detected in the wastewaters and surface waters of the United Kingdom [9]. Although a significant seasonal pattern was not found in the prevalence of foodborne viruses, virus contamination may not be neglected because a lot of water is used for fresh produce in green house during the winter. Therefore, the viral load and viability of irrigation water and reservoirs should be investigated in further study.”

40) L355-6 Reference is required.

  • This is not previous study but our observation during this study. Thus, there is no reference for this.

41) L358 Please, add ’in this study’ in the sentence.

  • We added ‘in this study’.

42) L362 Discussion on viral infectivity might be added in this paragraph.

  • We rewrote the paragraph from L395-400

43) L70-4 Please, rewrite these sentences for clarity. Try to avoid repetitions.

  • Line number need to be confirmed.

44) L383 Add citation (reference) for Guber et al.

  • We added citation for Guber et al.in L424
  • Guber AK, Williams DM, Quinn ACD, Tamrakar SB, Porter WF, Rose JB. 2016. Model of pathogen transmission between livestock and white-tailed deer in fragmented agricultural and forest landscapes. Environmental modelling & software, 80, 185-200.

45) L385-388 These sentences should be rewritten for clarity.

  • Thank you for your recommendation. Brief rephrase and additional description added in L423 -426

46) L392 ’confirmed’ is quite strong expression. Please, modify.

  • Thank you for your recommendation. We revised ’confirmed’ to ‘suggested’.

Reviewer 2 Report

The manuscript entitled “Effects of weather and environmental factors on the seasonal prevalence of foodborne viruses in irrigation waters in Gyeonggi province, Korea” by Wang and colleagues reports on the presence of enteric viruses in surface water in the korean region of Gyeonggi.

The study design is consistent with the aim of the study. The manuscript is structured, well written and the methods are described clearly. Presented results are of interest and the correlation analyses between viral detection rates and environmental factors enriched the significance of content. However, the inclusion of a higher number of samples could have improved the soundness of the study.

The study fits well the topic presented into the special issue “Enteric Virus Detection: Recent Developments and Application in Food and Waters”, so it may be of interest to include this manuscript in this collection. https://www.mdpi.com/journal/microorganisms/special_issues/enteric_virus_food_water.

Some minor issues, hereafter specified, should be clarified before publication.

L24-25, please specify the average viral titers. They should be presented in exponential format as it benefits the comparison made with other study in the discussion (L298-308).

L28-30. Please consider to rewrite the sentence for clarity.

L90; L183-184; L208; L228. Please provide the reference.

L91. Please specify the total number of samples (n=?).

L96. Space needed.

L105. Figure 1. Specify the sites on the map according with the letters used in Table 2.

L123. Please include details on quantification. They should be presented according to MIQE guidelines (Bustin et al., The MIQE Guidelines: Minimum Information for Publication of Quantitative Real-Time PCR Experiments, Clinical Chemistry, Volume 55, Issue 4, 1 April 2009, Pages 611–622, https://doi.org/10.1373/clinchem.2008.112797).

Also, specify if any correction of quantified titers has been made taking into account any quality/experimental parameter (e.g., process control recovery, extraction efficiency, concentrated volume). Moreover, process control recovery should be presented in the Results section.

L180-181. Comparing with Table 3, it is not clear if Sites A and C tested positive for more than 2 viruses. Please revise accordingly.

L212. Figure 3. Is the number of samples reported for each season (e.g., 31/76) or for the overall number of processed samples?

L216. Human norovirus GII have found to be high prevalent in spring. This is somehow different compared to the common norovirus seasonality. Do the author have any explanation of that? A discussion focusing on this point should be provided in the discussion section.

L298-308. Please check the format of the exponential titers reported. Also, specify the average viral titers in exponential format (L298-300) as it would be beneficial for the comparison.

L405-407. Please move the funding statement from “Acknowledgement” to the “Funding” section.

Author Response

1) L24-25, please specify the average viral titers. They should be presented in exponential format as it benefits the comparison made with other study in the discussion (L298-308).

  • Thank you for your critical review. We added the average viral loads in the abstract. As the average viral load is low, the form of (e.g.) -1.48 log 10 genome/copies is more suitable than 10-1.48 genome/copies in this study. Also, we rewrote L328-334, accordingly.

2) L28-30. Please consider to rewrite the sentence for clarity. 

  • We revised this sentence accordingly in L28-30.

3) L90; L183-184; L208; L228. Please provide the reference.

  • Thank you for your critical review. Figures were cited in this Lines: L92, L210, L236, 256

4) L91. Please specify the total number of samples (n=?). 

  • The total number of samples was revised in L98

5) L105. Figure 1. Specify the sites on the map according with the letters used in Table 2. 

  • Thank you for your recommendation. We rewrote the paragraph from L90-97 for clarity on the sampling sites. (revise Fig 1 with large indication)

6) L123. Please include details on quantification. They should be presented according to MIQE guidelines (Bustin et al., The MIQE Guidelines: Minimum Information for Publication of Quantitative Real-Time PCR Experiments, Clinical Chemistry, Volume 55, Issue 4, 1 April 2009, Pages 611–622, https://doi.org/10.1373/clinchem.2008.112797). Also, specify if any correction of quantified titers has been made taking into account any quality/experimental parameter (e.g., process control recovery, extraction efficiency, concentrated volume). Moreover, process control recovery should be presented in the Results section. . 

  • As reviewer #1 and 2 pointed this point, quantification method was added in the methods from L143-147. And result of recovery efficiency was added in L208

7) L180-181. Comparing with Table 3, it is not clear if Sites A and C tested positive for more than 2 viruses. Please revise accordingly. 

  • Thank you for your critical review. We rewrote the sentence for clarity as follows:
  • “The detection rate of virus-positive water samples for sites E, F, G, I, and J were 33.3%. Water sample was twice positive for virus in Site A and C, and only once positive in site D.”

8) L212. Figure 3. Is the number of samples reported for each season (e.g., 31/76) or for the overall number of processed samples?

  • We respect your comment on this point. Numerator represents a number of detected virus and male-specific coliphages for each season; denominator represents a total number of viruses and male-specific coliphages. Therefore, we revised the legend of Figure 3.

9) L216. Human norovirus GII have found to be high prevalent in spring. This is somehow different compared to the common norovirus seasonality. Do the author have any explanation of that? A discussion focusing on this point should be provided in the discussion section. 

  • We accept your critical review. As we know, the norovirus outbreak frequently occurs in winter. However, the interesting point in this study is that the virus contamination in reservoirs is associated with growing season such as spring to fall. Therefore, we discussed this part in L403-408 as follows.
  • “Depending on the conditions, foodborne viruses can survive in different environments for several months [5, 22]. Several weather factors, including rainfall, drought, humidity, temperature, and sunlight, affect the microbial infectivity [22]. Especially, low temperature, 10-66% relative humidity, and rainfall are closely associated with HuNoV outbreaks [66]. As Korean spring and autumn has low temperature, low relative humidity, and moderate rainfall, such an environmental factors may explain that HuNoV was frequently detected in these seasons rather than in summer.”

10) L298-308. Please check the format of the exponential titers reported. Also, specify the average viral titers in exponential format (L298-300) as it would be beneficial for the comparison.

  • We absolutely agree with reviewer. The exponential titer was changed into log titer. This paragraph was revised thoroughlty as follows.
  • “A previous study reported that the average viral loads of HuNoV was 2.30 log10 genome copies/l in a river in the Netherlands [45]. Similarly, the viral load of AdenoV ranged from 2.79 to 2.93 log10 genome copies/l in a river in Taiwan [21]. Furthermore, RotaV concentrations were as high as 14 log10 genome copies/l in surface river water in Germany [47]. However, in this study, the viral loads of viruses were low (−1.48 to 1.55 log10 genome copies/l). These results suggest that the reservoirs and river examined in this study indicated significantly lower viral contamination than water sources tested in previous studies.”

11) L405-407. Please move the funding statement from “Acknowledgement” to the “Funding” section. 

  • Thank you for your critical review. ‘This work was supported by the Basic Science Research Program through the National Research Foundation of Korea (NRF) funded by the Ministry of Education (NRF2018R1A6A1A03025159) ‘ was moved to the ‘Funding’ section

Reviewer 3 Report

Manuscript. No.: #874350

Title: Effects of weather and environmental factors on the seasonal prevalence of foodborne viruses in irrigation waters in Gyeonggi province, Korea

Authors: Zhaoqi Wang, Hansaem Shin, Soontag Jung, Daseul Yeo, Hyunkyung Park, Sangah Shin, Dong Joo Seo, Ki Hwan Park, Changsun Choi

 Overall, the manuscript is well written and detailed - clear, professional and scientific English language is used throughout. The study’s experimental design is well constructed and provides interesting data on microbial water quality standards in different sources within the sites studied. Such datasets are valuable in advancing our understanding of microbial water quality conditions in relation to climate and environmental conditions and will be very useful to better future water quality regulations and land management practices.

 Please see comments/ suggestions below for the improvement of the manuscript.

Line 49: Please provide a citation here.

Line 101: please provide manufacturer detail of the virosorb cartridge filter.

Methods, Section 2.1: How were the samples transported to the lab? Were water samples filtered on site? If so, how were filters transported? Based on the methods of coliform enumeration, authors must have transported the actual water to the lab. How was that done? Please add these details here under methods.

Figure 1: This map is good to have here, but I strongly suggest a better quality of the map, at this point it’s not very clear.

Line 118: Please provide the vivaspin manufacturer information here.

Lines 118-119: Did you take the filtrate from the 0.45u filtration and then do the concentration step? This is what it seems like considering the fact that these are viral samples. Please make sure this is clearly mentioned in this section.

  • Please make sure italicize coli or Escherichia coli in all instances across the manuscript. Some examples would be lines 136, 139, 174, 175.

Methods, section 2.4: Authors mention E. coli counts in results and tables. How was E. coli specifically enumerated? This needs to be detailed in the methods section clearly. What was the time between sample collection and bacterial, coliform and E. coli enumerations? How were controls (positive and negative) evaluated? Was E. coli confirmation done based on EPA standards? As this is one of the key experimental design methods, these need to be detailed a lot more. Simply mentioning a EPA method is not enough.

Figures 2, 3, 4: Better quality image required. At this point the writings and graphical representation is unclear.

Author Response

1) Line 49: Please provide a citation here. 

  • A news report of severe drought was cited in this sentence in L51.

2) Line 101: please provide manufacturer detail of the virosorb cartridge filter.

  • Thank you for your critical review. The company name was added in L108.

3) Methods, Section 2.1: How were the samples transported to the lab? Were water samples filtered on site? If so, how were filters transported? Based on the methods of coliform enumeration, authors must have transported the actual water to the lab. How was that done? Please add these details here under methods 

  • Thank you for your critical review. In order to minimize the microbial growth during the sampling process, we used the rental car and ice box. All samples were transferred directly to laboratory within 2-3 hours. Virus elution and concentration was performed on the same day.
  • For detecting indicator microorganisms, the method was added from L110-112.

4) Figure 1: This map is good to have here, but I strongly suggest a better quality of the map, at this point it’s not very clear.

  • Thank you for your recommendation. As reviewer #2 and #3 mentioned this point, we revise Figure 1.

5) Line 118: Please provide the vivaspin manufacturer information here..

  • Thank you for your critical review. Vivaspin manufacturer information was cited in L131.

6) Lines 118-119: Did you take the filtrate from the 0.45u filtration and then do the concentration step? This is what it seems like considering the fact that these are viral samples. Please make sure this is clearly mentioned in this section.. 

  • Thank you for your critical review. Yes, it’s the concentration step for the filtration of the microorganism which size was big than 0.45 μ We revised this sentence from L128-130 for clarity.

7) Please make sure italicize coli or Escherichia coli in all instances across the manuscript. Some examples would be lines 136, 139, 174, 175.. 

  • Thank you for your critical review. We italicized all coli and Escherichia coli thoroughly.

8) Methods, section 2.4: Authors mention E. coli counts in results and tables. How was E. coli specifically enumerated? This needs to be detailed in the methods section clearly. What was the time between sample collection and bacterial, coliform and E. coli enumerations? How were controls (positive and negative) evaluated? Was E. coli confirmation done based on EPA standards? As this is one of the key experimental design methods, these need to be detailed a lot more. Simply mentioning a EPA method is not enough.. 

  • Thank you for your critical review. The method of coli was specifically enumerated from L162-167

9) Figures 2, 3, 4: Better quality image required. At this point the writings and graphical representation is unclear.. 

  • Thank you for your critical review. All figures with high resolution were replaced in the manuscript. However, pdf version for reviewers may seem to be unclear compared with original file. Please understand this.
